# OpenReview forum: "Benchmarking LLMs on Safety Issues in Scientific Labs"
_ICLR.cc/2025/Conference — Submitted to ICLR 2025_

### Official Review · Reviewer_GBeA · 2024-10-29

**Soundness:** 1
**Presentation:** 2
**Contribution:** 2
**Rating:** 3
**Confidence:** 4

**Summary:**

This paper argues that although there is much previous work on safety, trustworthiness, truthfulness, fairness and privacy for LLMs, no previous work has investigated lab safety specifically, which involves a different aspect of trustworthiness. The paper develops a benchmark to evaluate LLMs in terms of lab safety where they develop a taxonomy based on OSHA protocols, curate a set of questions based on the taxonomy, and then evaluate a set of foundation models and humans for their accuracy in selecting the correct answer to the questions.

**Strengths:**

The problem is well motivated and justified, and seems important considering the use of LLMs in many safety-critical areas. The lab safety taxonomy based on various OSHA requirements seems helpful and useful for benchmarking. The evaluation has good diversity of LLM models, with 17 foundation models, including both open source, proprietary and scientific models and different prompting techniques. Finally, I appreciate that the code is released open source.

**Weaknesses:**

I have some issues with the evaluation. Starting with the human evaluation (see question 1): the paper mentions they used individuals who had undergone lab safety training but what exactly does that entail? I am wondering if this was a sufficiently fair comparison-- it may be the case that the humans were being tested on very specific lab safety questions that they had not encountered or been trained for (e.g., lab safety training was for a broad range of biology wet lab safety questions when an individual only works in or is only trained in a specific sub field of biology safety or vice versa). Moreover, I would argue the LLM in this scenario should be considered an expert, and therefore should be compared to only human "experts" (i.e., not junior scientists like undergraduates).

In addition, the authors mention the goal is to determine whether or not LLMs can be trusted to be more reliable than humans in decision making and planning for lab safety. Can the authors comment on the applicability of using the benchmark multiple-choice question & answer as a proxy for evaluating this (see question 2)? In practice, the way a user engages with the LLM for these types of decisions is very different (i.e., they are probably not going to ask it a multiple choice question with the answers listed.) I am wondering if this is really a fair methodology in order to evaluate how well LLMs follow lab safety protocols.

**Questions:**

(1a) For the human evaluation, how were the human participants selected and to what extent did they have direct experience or training related to the specific questions being asked in the evaluation?
(1b) Why did you include junior scientists in the comparison?

(2a) Why were multiple choice questions chosen as the evaluation technique?
(2b) Can the authors comment on the applicability of using the benchmark multiple-choice question & answer as a proxy for evaluating this?

---

> ### Author Response · Authors · 2024-11-21
> **Reponse to Question 1 and one concern in Weakness 1**
>
> We deeply appreciate the reviewer’s insightful feedback and profound observations, which have been incredibly inspiring for our work. Your thoughtful comments have prompted us to refine and strengthen our approach. In particular, your suggestions regarding the selection of human participants have provided us with valuable perspectives and prompted us to consider critical aspects that we had not previously emphasized. Thank you for your invaluable contributions to improving the clarity and depth of our study.
>
> ### **Question 1a**: For the human evaluation, how were the human participants selected and to what extent did they have direct experience or training related to the specific questions being asked in the evaluation?
>
> **Response**: Thank you for raising this question. Below, we provide details about the selection process and qualifications of our human participants, as well as steps for improving future evaluations.
>
> **1. Selection of Participants**
> Our participants were primarily from a large research-oriented university. The group included **35 graduate-level or higher-education individuals** and **15 undergraduates**. The table below displays the distribution of undergraduate and graduate students for each questionnaire used in the human evaluation. All participants had prior laboratory experience. **Undergraduates were required to have at least one year of lab experience**, while **graduate students and higher-education participants needed at least three years of lab experience.**  All participants had **completed lab safety training** relevant to their specific academic disciplines. Participants were initially categorized based on their **academic disciplines** to align their expertise with the general focus of the evaluation. However, we acknowledge that this was a relatively broad classification and did not include finer distinctions within subfields.
>
>
> |  | Physics | Chemistry | Biology | General |
> |---|---|---|---|---|
> | Undergraduate | 3 | 3 | 5 | 4 |
> | Graduate or higher | 5 | 7 | 12 | 11 |
>
>
> ### **One Concern in Weakness 1**: it may be the case that the humans were being tested on very specific lab safety questions that they had not encountered or been trained for (e.g., lab safety training was for a broad range of biology wet lab safety questions when an individual only works in or is only trained in a specific sub field of biology safety or vice versa).
>
> **Response: Students are trained more than they need**
>
> We conducted a survey at the same large research university, involving six graduate students with over three years of laboratory experience from six labs across three different disciplines. All participants indicated that they had passed the basic lab safety training for their respective disciplines. While they were more familiar with safety protocols specific to their own labs, they also acknowledged that the training covered a wide range of topics beyond those directly relevant to their lab work. Additionally, the students noted that while they did not know all the answers to our test questions, the knowledge points were not entirely unfamiliar to them.
>
> ### **Question 1b**: Why did you include junior scientists in the comparison?
>
> **Response**: Thank you for your question regarding the inclusion of junior scientists in our evaluation. Below, we outline the rationale for including undergraduate participants and our plans for refining future analyses.
>
> 1. We included undergraduate participants because **they are also active in laboratory environments and face the same safety challenges** as more experienced researchers. Their inclusion provides insights into how less-experienced individuals approach lab safety problems, which is essential for understanding the full spectrum of safety-related knowledge gaps.
>
> 2. By including both undergraduate and graduate-level participants, we aimed to explore **how lab safety knowledge varies across different levels of experience**. We used laboratory experience as a proxy for knowledge level and compared the performance of these groups to identify the impact of experience on lab safety understanding.
>
> 3. The table below shows the accuracy of undergraduate students or Graduate or higher degree students on sampled LabSafety Bench on four questionnaires. We observed that **undergraduates generally performed worse in the evaluation**, which highlights the importance of considering expertise when interpreting results. In future analyses, we plan to separate the performance statistics for undergraduates and graduate-level participants to provide a clearer picture of how expertise impacts safety knowledge.
>
> |  | Physics | Chemistry | Biology | General |
> |---|---|---|---|---|
> | Undergraduate | 45.33 | 58.67 | 61.6 | 61 |
> | Graduate or higher | 60 | 70.29 | 70.33 | 70.55 |

---

> ### Author Response · Authors · 2024-11-21
> **Response to Question 2 and one concern in Weakness 2**
>
> ### **Question 2a**: Why were multiple choice questions chosen as the evaluation technique? (2b) Can the authors comment on the applicability of using the benchmark multiple-choice question & answer as a proxy for evaluating this?
>
> **Response**: We choose multi-choice questions as they support the usage of **standardized and quantifiable metrics like accuracy, making it feasible to compare the performance of different LLMs**. This is similar to the initial studies in MMLU [1], which first proposed benchmarks for Massive Multitask Language Understanding and Massive Multi-discipline Multimodal Understanding and Reasoning. It uses multiple-choice questions to facilitate the evaluation.
> We fully agree that multiple-choice questions alone cannot comprehensively capture the risks associated with lab safety. To address this limitation, we are actively working on exploring open-ended scenarios and formats to expand the evaluation dimensions. However, despite the trade-off in format for the sake of measurability, our questions are designed to **cover a broad range of knowledge domains**. These are grounded in **authoritative sources** such as WHO [2] and OSHA [3]. Moreover, the questions **span various difficulty levels** and **encompass biology, chemistry, and physics disciplines**. Although these questions are “not complex”, there are some notable gaps between different LLMs, and even the best LLM (GPT-4o) only scored 86.27% on both text-only and text-with-image questions in LabSafety Bench and 82.35% on hard questions, demonstrating that this benchmark is still challenging to current LLMs.
>
> Importantly, all questions have been **reviewed and refined by human experts** to ensure that **the options are non-trivial and effectively test the LLMs' understanding of the underlying knowledge points**.
>
>
> ### **Question 2b**: Lab safety tasks are highly knowledge-intensive and involve domain-specific, professional-level questions.
>
> **Response**: Evaluating free-form answers for such tasks poses significant challenges. There is currently no reliable model or tool capable of accurately assessing the correctness of free-form responses for such specialized topics. Methods like keyword matching or scoring are prone to errors and cannot ensure accurate judgments. In contrast, multi-choice questions allow human experts to rigorously validate the correctness of each question and its options. This is the key reason why we use multi-choice questions rather than answers as a proxy for evaluation.
>
> ### **A Concern in Weakness 2**: In practice, the way a user engages with the LLM for these types of decisions is very different (i.e., they are probably not going to ask it a multiple-choice question with the answers listed.) I am wondering if this is really a fair methodology in order to evaluate how well LLMs follow lab safety protocols.
>
> **Response**: We acknowledge the difference between real-world user interactions with LLMs and the structured multiple-choice format used in our evaluation. However, **every lab safety hazard, no matter how complex, ultimately stems from fundamental safety knowledge points**. For instance, the ethanol pipeline explosion mentioned in the introduction underscores the critical principle that flammable liquids must be handled to avoid ignition from static electricity during transfer. In our benchmark, such hazards are addressed through multiple targeted questions about the handling of flammable substances. **By assessing LLMs on these individual safety knowledge points, our methodology ensures a thorough evaluation of their understanding of critical lab safety protocols, even if the interaction format differs.**
>
>
> Thank you once again for your valuable and constructive feedback. If there are any aspects that remain unclear or if you have further questions, we would be glad to provide additional clarification. We hope our responses have addressed your concerns and would greatly appreciate it if you could consider revisiting and potentially adjusting your score.
>
> Best regards,
>
> Authors of Submission 12083
>
> [1] Hendrycks, Dan, et al. "Measuring massive multitask language understanding." arXiv preprint arXiv:2009.03300 (2020).
>
> [2] WHO.Laboratorybiosafetymanual.WHO,5(2):1–109,2003.
>
> [3] OSHA. Laboratory safety guidance. URL https://www.osha.gov/sites/ default/files/publications/OSHA3404laboratory-safety-guidance. Pdf

---

> > ### Comment · Reviewer_GBeA · 2024-11-25
> > **Response to Authors**
> >
> > I thank the authors for their response.
> >
> > Question 1/Weakness 1:
> > The authors mention that undergraduates and junior scientists were included to “provides insights into how less-experienced individuals approach lab safety problems” but isn’t the point of this study to evaluate LLMs in lab settings? I still stand by my original point that I think only senior scientists and experts should be compared with the LLMs because they should be treated as an expert level system. Moreover, I do not find it fair that the humans are being evaluated on questions outside their particular discipline. I think it would make more sense to only compare with field specific questions for the human experts.
> >
> > Weakness 2/ Question 2:
> > I recognize that multiple choice questions are a standardized way for evaluation, however I still think there are better ways to evaluate lab safety, particularly since the way users engage with an LLM for these types of scenarios would not be using multiple choice questions.
> >
> > Based on the above and additional important points brought up by the other reviewers I feel this work has some fundamental technical issues (for me particularly in the evaluation, but I agree with other reviewers issues with framing of the motivation as well). I reduce my score.

---

> > > ### Author Response · Authors · 2024-12-01
> > >
> > > We appreciate the reviewer’s feedback and have carefully considered the concerns raised. While we respect the reviewer’s perspective, we respectfully disagree on certain points. Below, we address the specific concerns:
> > > ### **Question 1 / Weakness 1**: Mixing Results of Junior and Senior Researchers in Human Evaluation, **and** Humans Shouldn’t be Evaluated on Questions outside their Particular Discipline
> > > We have accepted part of the reviewer’s suggestion that the results from junior scientists, such as undergraduates, and senior scientists, such as researchers with over three years of lab experience, should not be combined in human evaluation. To address this, **we have separated the results of junior and senior scientists** in the initial rebuttal. The results show that senior scientists outperform junior scientists by approximately 10% on average.
> > >
> > > To further clarify the comparison with human experts, we selected the **top-3 performers** from all questionnaires (all of whom were senior researchers) and used their results as a proxy for human expert performance. This was already included in the original paper, where **top-3 human performance** was compared to LLMs in **Figure 5** and **Table 2**. We believe this adequately addresses the reviewer’s concerns by clearly presenting the performance of human experts on sampled LabSafety Bench questions.
> > >
> > > At the same time, we believe including the performance of junior researchers is equally important. While LabSafety Bench is designed to evaluate LLMs’ lab safety awareness, it can also measure human lab safety awareness. **For undergraduates, who are actively conducting experiments, understanding their gaps in lab safety awareness further supports our motivation**: even individuals who have undergone safety training may still lack sufficient awareness and seek help from LLMs.
> > >
> > > Additionally, as mentioned in our previous rebuttal, we consulted **six lab workers from six labs in a large research university. They consistently reported receiving broader safety training than required for their current experiments because many experiments share overlapping protocols in one discipline.** For example, a lab focused on research area A may still train its members in area B protocols to prepare for potential future experiments. This reflects the reality of modern lab safety training. Thus, we only divide the questions based on chemistry, physics, or biology disciplines. Therefore, we respectfully disagree with the reviewer’s assertion, made without evidence, that all laboratory personnel should only receive safety training relevant to their immediate experiments.
> > >
> > > Finally, in sampling questions for the survey, we deliberately excluded the most specific questions, such as those related to BSL-3/4 protocols, as this knowledge is only required for individuals working in such facilities. While these topics are included in the full LabSafety Bench, they were not sampled in the human evaluation to ensure relevance.
> > >
> > > ### **Question 2 / Weakness 2**: Use of Multiple-Choice Questions Instead of Free-Form Questions
> > >
> > > We respectfully disagree with the reviewer’s perspective on this point. Lab safety tasks are inherently knowledge-intensive, and even with multiple-choice questions, the highest performance achieved by LLMs is only 86%. In this context, we do not believe that LLMs can accurately evaluate free-form answers. **Currently, no model—or any method independent of expert intervention—can reliably evaluate whether LLM-generated answers to free-form questions are safe.**
> > >
> > > After thorough deliberation, we determined that multiple-choice questions are the most suitable format for evaluating lab safety awareness. While the reviewer suggested that free-form questions might better reflect real-world use cases, the reality is that users are already using LLMs for lab safety-related queries without any benchmark to assess LLMs’ safety awareness. Free-form evaluations would be unreliable given the current performance limitations of LLMs, as no system can achieve the necessary accuracy to evaluate free-form responses reliably. Furthermore, relying on experts to manually assess every LLM-generated answer would be impractical and troublesome and introduce additional biases during evaluation. Thus, we believe that multiple-choice questions currently offer the most effective and scalable approach to evaluate LLMs' lab safety awareness.

---

### Official Review · Reviewer_hEE9 · 2024-11-02

**Soundness:** 3
**Presentation:** 3
**Contribution:** 2
**Rating:** 5
**Confidence:** 3

**Summary:**

This paper explores the performance of LLMs when they are applied to safety issues in scientific labs. The author(s) identifies a new taxonomy aligned with Occupational Safety and Health Administration protocols and based on it, they develop a benchmark for LLM-based lab safety. The benchmark includes 765 multiple-choice questions that human experts manually verify. A systematic evaluation approach is defined to evaluate the performance of several SOTA LLMs such as GTP-4o. Several findings are drawn from the evaluation, such as that LLMs can outperform human beings in lab safety but are still prone to critical errors or safety crises.

**Strengths:**

1. A sound study with carefully designed experiments and evaluations. I must acknowledge the authors' efforts to build corpora in this specific domain. I understand that it needs a huge amount of human effort to compose and examine the data for a high-quality corpus. Particularly, lab safety in different domains is very domain-specific and requires a lot of domain knowledge. A benchmark is highly valuable in developing useful LLMs for the purpose of safety guidance.

2. Good and clear presentation. The paper has a clear motivation, methodology, and evaluation results. The findings from the study are convincing. It is interesting to see that CoT and few-shot learning have minimal impact on LLM performance. From the evaluation, the authors also identify that most LLMs have difficulties in Radiation Hazards, Physical Hazards, Equipment Usage, and Electricity Safety.

**Weaknesses:**

1. Lack of insights. The findings from the work are not surprising. For instance, the authors find that LLMs are not fully reliable in the application of lab safety. This result is apparent. At present, in any safety-critical domain, a big concern with LLMs is their safety or reliability. Compared with such findings, potential solutions would be more valuable, even in this specific domain.

2. The authors spend several pages discussing the comparison of several mainstream LLMs when they are applied to lab safety. I couldn't see the meaning of the comparison results. I understand that the comparison can reflect which LLM performs better than the other. But what does that imply? LLM competition develops fast. The results might be obsolete quickly. I do not think the comparison is necessary or important.

3. Lack of evaluation of benchmarks. The authors claim their contribution is the benchmark for lab safety applications. If so, the benchmark shall be carefully evaluated to convince the followers the benchmark is good enough to train LLMs or investigate new approaches to achieve more accurate lab safety guidance using LLMs or other LLM-assisted approaches.

**Questions:**

1. Could you explain the necessity of comparing LLM performance for the benchmark?
2. How could we assess the quality of the benchmark?
3. Besides data, what are the real challenges when LLMs are applied to the lab safety problem?

---

> ### Author Response · Authors · 2024-11-21
> **Response to Weakness 1 and 2 & Question 1**
>
> Dear Reviewer hEE9,
>
> We sincerely thank the reviewer for their thorough reading and for providing us with valuable feedback. The insights, especially those prompting us to more clearly articulate the high quality of our benchmark, have been immensely helpful, and we greatly appreciate your thoughtful questions and suggestions.
>
> ### **Weakness 1**: Lack of insights. The findings from the work are not surprising. For instance, the authors find that LLMs are not fully reliable in the application of lab safety. This result is apparent. At present, in any safety-critical domain, a big concern with LLMs is their safety or reliability. Compared with such findings, potential solutions would be more valuable, even in this specific domain.
>
> **Response**:
> We respectfully disagree with the criticism regarding a supposed “lack of insights” in our work. While it is widely acknowledged that LLMs are not fully reliable, and it is indeed evident that current models cannot achieve 100% accuracy on the LabSafety Bench, we believe that evaluating their performance on lab safety-related tasks remains both meaningful and necessary. This study, to our knowledge, is the **first work that benchmarks the capabilities of LLMs specifically in the context of lab safety**.
> Moreover, **the evaluation results do reveal several valuable findings**. For instance, “Proprietary models are generally better at lab safety issues compared to open-weight models.”, “CoT and few-shot learning have minimal impact on performance but hints significantly boost the performance of smaller open-weight models.”  and “The LLM demonstrates relatively weaker performance on physics-related safety questions compared to other disciplines.”
>
> ### **Weakness 2**: The authors spend several pages discussing the comparison of several mainstream LLMs when they are applied to lab safety. I couldn't see the meaning of the comparison results. I understand that the comparison can reflect which LLM performs better than the other. But what does that imply? LLM competition develops fast. The results might be obsolete quickly. I do not think the comparison is necessary or important. **& Question 1**: Could you explain the necessity of comparing LLM performance for the benchmark?
>
> **Response**: We appreciate the reviewer’s concern regarding the relevance of comparing multiple LLMs, especially given the rapid pace of model development. However, we believe **such comparisons remain crucial and valuable** for several reasons. **First**, evaluating mainstream LLMs on lab safety tasks helps **identify current performance limits and gaps**, clarifying which models are better suited for safety-critical applications and where improvements are needed, as is **standard in benchmark studies** [1, 2, 3]. **Second**, in high-risk domains like lab safety, these evaluations **guide model selection by highlighting specific strengths and weaknesses**, enabling researchers to focus on targeted improvements. **Finally**, while models evolve quickly, our benchmark provides **a valuable baseline** to track progress over time and assess advancements in bridging the gap between benchmark performance and real-world safety needs.
>
> [1] Yue X, Ni Y, Zhang K, et al. Mmmu: A massive multi-discipline multimodal understanding and reasoning benchmark for expert agi[C]//Proceedings of the IEEE/CVF Conference on Computer Vision and Pattern Recognition. 2024: 9556-9567.
>
> [2] Hendrycks D, Burns C, Basart S, et al. Measuring massive multitask language understanding[J]. arXiv preprint arXiv:2009.03300, 2020.
>
> [3] Sun L, Huang Y, Wang H, et al. Trustllm: Trustworthiness in large language models[J]. arXiv preprint arXiv:2401.05561, 2024.

---

> > ### Comment · Reviewer_hEE9 · 2024-11-23
> >
> > The authors claimed that that is the first work that benchmarks the capabilities of LLMs specifically in the context of lab safety. I agree with that benchmark is valuable and worth doing. My point is that if the authors emphasized the benchmark, they should have evaluated the benchmark but not the LLMs. That is, other researchers need to understand why they should  follow your benchmark in this field. However, the authors didn't convince me the quality of the benchmark.
> >
> > The authors claimed some findings from the experiments. These findings are important, but still I don't think they are profound. When applying LLMs to Lab safety, I believe there must be some questions that are more important than those addressed in the paper. For instance, to what extent LLMs can provide assistances or replace human beings in this task? How do human beings interact with LLMs for safety requirements in labs?

---

> > > ### Author Response · Authors · 2024-11-26
> > > **Key Insights, Potential Applications, and Broader Implications**
> > >
> > > ### **Question**: The authors claimed some findings from the experiments. These findings are important, but still I don't think they are profound. When applying LLMs to Lab safety, I believe there must be some questions that are more important than those addressed in the paper. For instance, to what extent LLMs can provide assistances or replace human beings in this task? How do human beings interact with LLMs for safety requirements in labs?
> > >
> > > ### **Response**:
> > > **Key Findings of Our Study**
> > >
> > > Our study highlights that current LLMs are not yet reliable enough to be directly used for addressing lab safety-related questions. Given the high stakes involved in laboratory safety, where improper handling of safety issues can lead to severe accidents, LLMs must achieve nearly 100% accuracy on lab safety benchmarks to be considered trustworthy. This is further emphasized by the fact that LLMs cannot take responsibility for the consequences of their generated answers.
> > >
> > > While GPT-4o achieved an accuracy of 86.27%, both the Item Characteristic Curve (ICC) analysis of its **Ability Level** and this accuracy level itself indicates that it is not yet suitable for direct use in lab safety-related tasks. Moreover, as shown in **Table 2**, GPT-4o achieved only 90% accuracy even within the best-performing subcategory, demonstrating that there is significant room for improvement across all categories.
> > >
> > >
> > > **How LLMs Can Provide Safe Assistance**
> > >
> > > Despite these limitations, GPT-4o’s relatively strong performance across most subcategories compared to other LLMs makes it a potential candidate for assisting with lab safety in a supervised and controlled manner. In our case studies, we observed that while GPT-4o may misjudge the priority of certain hazards, its CoT analysis reveals an understanding of which options are dangerous. Furthermore, **GPT-4o performed significantly better in identifying the safest option** rather than simply determining which options are hazardous.
> > >
> > > This suggests that **GPT-4o could be effectively utilized for tasks focused on identifying the safest course of action**. For example, instead of being used directly by experimenters, it would be safer and more effective for a **lab manager**—who possesses robust safety awareness—to query the LLM for relevant safety details before an experiment. The lab manager could then validate and relay this information to experimenters, ensuring safety while saving time on manual research. We have expanded on this point in Appendix D of our paper.
> > >
> > > **Contribution of Our Work: Drawing Parallels to Early Studies of LLM for Medicine**
> > >
> > > Lab safety, like medicine, is a high-risk domain where errors can have serious consequences. Early studies evaluating the application of LLMs in medicine, such as one highly influential study with over 700 citations [1], took a similar approach to ours by focusing on performance in medical competency exams and benchmark datasets, demonstrating the value of identifying gaps in LLM capabilities and urging caution in their real-world application.
> > >
> > > Similarly, our work represents the **first study evaluating LLMs in the context of lab safety**, a critical and underexplored domain. By adopting a comparable evaluation approach, our study underscores the current limitations of LLMs in lab safety awareness while providing a structured framework for their assessment. Given the parallels to early influential studies in medicine, this work holds significant value in guiding the cautious and informed adoption of LLMs in high-stakes laboratory environments.
> > >
> > > [1] Nori H, King N, McKinney S M, et al. Capabilities of gpt-4 on medical challenge problems[J]. arXiv preprint arXiv:2303.13375, 2023.

---

> ### Author Response · Authors · 2024-11-21
> **Response to Weakness 3 and Question 2-3**
>
> ### **Weakness 3**: Lack of evaluation of benchmarks. The authors claim their contribution is the benchmark for lab safety applications. If so, the benchmark shall be carefully evaluated to convince the followers the benchmark is good enough to train LLMs or investigate new approaches to achieve more accurate lab safety guidance using LLMs or other LLM-assisted approaches.
> **& Question 2**: How could we assess the quality of the benchmark?
>
> **Response**: Thank you for raising this important point. We have implemented several measures to ensure the benchmark’s effectiveness and comprehensiveness. First, it **covers a broad range of lab safety knowledge points**, guided by the Risk Management and Safety team of a major U.S. university and OSHA protocols, ensuring alignment with real-world safety practices and standards like OSHA and WHO. Second, **each question underwent rigorous expert review to ensure clarity, reasonable options, unique answers, and appropriate difficulty**, making the benchmark a reliable tool for assessing LLMs in safety-critical tasks. Finally, while fine-tuning has limitations for knowledge-intensive tasks like lab safety [4], our curated dataset **lays the groundwork for future pretraining efforts, with plans to scale and validate its utility in enhancing LLM performance.**
>
> ### **Question 3**: Besides data, what are the real challenges when LLMs are applied to the lab safety problem?
>
> **Response**: Thank you for the insightful question. The real challenges depend on "how LLMs are applied" to the lab safety problem. Let’s explore three specific scenarios:
> **When LLMs are used to assist students in lab safety training**, they can act as supplementary resources to support student learning by offering interactive safety training modules. For example, they can provide explanations of safety protocols, or quiz students on critical procedures. However, LLMs might provide **incorrect or overly generalized answers, which can mislead students**.
> **When LLMs are used in autonomous self-driving labs [4,5,6,7]**, they can propose experimental workflows, predict outcomes, and so on. Meanwhile, they should identify potential safety issues. However, it is challenging for LLMs to **handle highly complex and interdependent systems in self-driving labs**. It is of course possible to set up a virtual environment before implementation in a physical LLM-empowered lab. However, the virtual lab environment must **accurately simulate real-world conditions**.
>
>
> Thank you again for your thoughtful and constructive feedback. If you have any remaining questions or need further clarification, we would be more than happy to assist. We hope our responses have addressed your concerns effectively, and we kindly ask you to consider revisiting and potentially adjusting your score.
>
> Best regards,
>
> Authors of Submission 12083
>
> [4] Daniil A. Boiko, Robert MacKnight, Ben Kline and Gabe Gomes. Autonomous chemical research with large language models.  Nature. volume 624, pages 570–578 (2023).
>
> [5] Milad Abolhasani and Eugenia Kumacheva. The rise of self-driving labs in chemical and materials sciences. Nature Synthesis. volume 2, pages 483–492 (2023)
>
> [6] Latif, Ehsan, Ramviyas Parasuraman, and Xiaoming Zhai. "PhysicsAssistant: An LLM-Powered Interactive Learning Robot for Physics Lab Investigations." arXiv preprint arXiv:2403.18721 (2024).
>
> [7] Gary Tom et al. Self-Driving Laboratories for Chemistry and Materials Science. Chem. Rev. 2024, 124, 16, 9633–9732.

---

> ### Author Response · Authors · 2024-11-26
> **Further Evaluation Supporting the High Quality of Our Benchmark**
>
> ### **Question**: My point is that if the authors emphasized the benchmark, they should have evaluated the benchmark but not the LLMs. That is, other researchers need to understand why they should follow your benchmark in this field. However, the authors didn't convince me the quality of the benchmark.
>
> ### **Response**:
>
> Thank you for the insightful feedback. In addition to the points highlighted in our previous rebuttal—namely, that the comprehensive data collection process forms the foundation of our dataset's high quality, and the human experts' panel reviews further ensure its exceptional reliability—we also conducted several statistical evaluations of LabSafety Bench to provide additional evidence of its high quality, detailed in Appendix B. Specifically:
> - **Appendix B.1** presents the **distribution of word counts** for our questions.
> - **Appendix B.2** outlines the **distribution of the number of categories per question**.
> - **Appendix B.3** provides the **category overlap statistical results**.
>
> Additionally, we **newly include some statistical evaluations in Appendix B.4 and B.5** in the revision. **Appendix B.4** demonstrates **the diversity of our dataset in the embedding space**, with results visualized in **Figure 11**. Finally, in **Appendix B.5**, we introduce **Item Response Theory (IRT)**, a psychometric model used to analyze the characteristics of test items and the abilities of test-takers.
>
> IRT evaluates each question using three key parameters: **discrimination**, which measures the ability of a question to distinguish between high- and low-ability test-takers; **difficulty**, which reflects the level of challenge the question poses; and **guessing**, which estimates the likelihood of a correct answer being selected through random guessing.
>
> The results in **Figure 12** demonstrate the high quality of our dataset. The strong discrimination ability of items ensures **reliable differentiation between varying levels of test-taker ability**, while **the well-balanced difficulty level presents an appropriate challenge**. Additionally, the low guessing parameter indicates that the **items are effectively designed to minimize random success**, contributing to the robustness of the assessment. Overall, these findings affirm that our dataset is well-constructed, providing an effective and reliable tool for evaluating test-taker abilities across a broad spectrum.
>
>
> Moreover, using IRT, we plotted the **Item Characteristic Curve** and mapped the accuracy of GPT-4o, the best-performing model. The results in **Figure 13** reveal that GPT-4o corresponds to an **ability level of 1.24**, which is significantly below the **level 3** required for near-complete mastery. This finding highlights the significant gap between the current performance of GPT-4o and the level of expertise necessary for comprehensive lab safety understanding.

---

### Official Review · Reviewer_t1YP · 2024-11-04

**Soundness:** 3
**Presentation:** 3
**Contribution:** 2
**Rating:** 3
**Confidence:** 4

**Summary:**

This paper introduces LabSafety Bench, a benchmark for evaluating LLM capabilities in understanding and providing guidance on laboratory safety protocols. The benchmark contains 765 multiple-choice questions covering various aspects of lab safety, including hazardous materials handling, emergency responses, and equipment usage. The authors evaluate 17 models on this benchmark and compare their performance against human experts.

**Strengths:**

- The work addresses an important topic as LLMs are increasingly being integrated into scientific workflows
- The benchmark is comprehensive, covering multiple aspects of lab safety and aligned with established OSHA protocols
- The evaluation is thorough, including both text-only and multimodal questions, and comparing against human performance

**Weaknesses:**

- My main concern is with the practical significance of the benchmark. The primary use case presented - students asking LLMs about lab safety - is itself problematic, as students should be consulting official protocols and trained personnel rather than LLMs. In practice, I doubt students will solely consult LLMs on safety-related questions. Is this the main use case?
- The benchmark may not effectively capture real-world safety risks, as actual laboratory environments have multiple safeguards and verification processes beyond just knowledge of safety protocols. Real-world failures like the ones listed in the introduction involve many overlapping systems, and it is unclear how this benchmark will help prevent them.
- Human evaluation does not allow participants to consult external materials. This is problematic as if in practice humans can get 100% on this benchmark they wouldn't need LLMs to help with lab safety questions

**Questions:**

Are there any potential biases to using GPT-4 as part of the benchmark curation process?

---

> ### Author Response · Authors · 2024-11-21
> **Response to Weakness 1 and 2**
>
> Dear Reviewer t1YP,
>
> We sincerely thank you for your thoughtful and insightful suggestions and questions, particularly regarding the motivation behind our work and the associated recommendations. Your feedback has been invaluable, and we incorporated the details discussed in the rebuttal into the paper to provide greater clarity and depth. Thank you again for your constructive comments.
>
> ### **Weakness 1**: My main concern is with the practical significance of the benchmark. The primary use case presented - students asking LLMs about lab safety - is itself problematic, as students should be consulting official protocols and trained personnel rather than LLMs. In practice, I doubt students will solely consult LLMs on safety-related questions. Is this the main use case?
>
> **Response**: **Firstly**, even individuals who have completed lab safety training often demonstrate gaps in their lab safety knowledge, which can contribute to laboratory accidents [1]. This lack of certainty regarding certain topics may lead students to place undue trust in the plausible-sounding answers generated by LLMs.
> **Secondly**, LLMs have been demonstrated to assist **remote and autonomous self-driving labs**, where LLMs are leveraged to draft experimental plans and conduct various operations in the workflow. There is also a growing trend of utilizing LLMs in **diverse research environments**, highlighting their potential to streamline processes, reduce manual effort, and enhance overall efficiency [1,2,3,4]. In this setting, LLMs' awareness of potential risks, and safety protocols becomes critical to ensure reliable and secure operation.
>
> ### **Weakness 2**: The benchmark may not effectively capture real-world safety risks, as actual laboratory environments have multiple safeguards and verification processes beyond just knowledge of safety protocols. Real-world failures like the ones listed in the introduction involve many overlapping systems, and it is unclear how this benchmark will help prevent them.
>
> **Response**: We acknowledge the complexity of real laboratory environments; however, every lab safety hazard, **no matter how intricate, ultimately arises from a combination of fundamental safety knowledge points**. For example, the ethanol pipeline explosion mentioned in the introduction highlights the core principle that flammable liquids must be handled to avoid ignition from static electricity during transfer. In our benchmark, this specific hazard is addressed through multiple questions targeting the handling of flammable substances. By evaluating LLMs’ understanding of individual safety knowledge points, we aim to ensure comprehensive coverage of critical lab safety protocols.
>
> [1] Daniil A. Boiko, Robert MacKnight, Ben Kline and Gabe Gomes. Autonomous chemical research with large language models.  Nature. volume 624, pages 570–578 (2023).
>
> [2] Milad Abolhasani and Eugenia Kumacheva. The rise of self-driving labs in chemical and materials sciences. Nature Synthesis. volume 2, pages 483–492 (2023)
>
> [3] Latif, Ehsan, Ramviyas Parasuraman, and Xiaoming Zhai. "PhysicsAssistant: An LLM-Powered Interactive Learning Robot for Physics Lab Investigations." arXiv preprint arXiv:2403.18721 (2024).
>
> [4] Gary Tom et al. Self-Driving Laboratories for Chemistry and Materials Science. Chem. Rev. 2024, 124, 16, 9633–9732.

---

> ### Author Response · Authors · 2024-11-21
> **Response to Weakness 3 and Question 1**
>
> ### **Weakness 3**: Human evaluation does not allow participants to consult external materials. This is problematic as if in practice humans can get 100% on this benchmark they wouldn't need LLMs to help with lab safety questions
>
> **Response**: We would like to justify this setting with two key considerations:
>
> 1)  Since the goal was to assess the understanding and knowledge retention of researchers who had completed safety training and possessed significant lab experience, restricting external resources during human evaluation is essential to **accurately measure their baseline knowledge and independent problem-solving capabilities**.
>
> 2) In actual lab environments, humans are, of course, free to use any external resources. However, in our evaluation, **if humans are allowed to use external resources, it is highly possible that they use LLMs** (numerous recent studies and reports have highlighted, that STEM education and research increasingly involve leveraging LLMs for problem-solving and learning enhancement [5, 6]). In this way,  our evaluation would turn into a test of LLMs versus LLMs, undermining the purpose of the comparison
>
>
>
> ### **Question 1**: Are there any potential biases to using GPT-4 as part of the benchmark curation process?
>
> **Response**: We acknowledge that using GPT-4 to generate questions and incorrect options may introduce some bias. However, we have implemented various measures to mitigate its impact. After refinement, some modified options were found to be inaccurately phrased, irrelevant to the question, or transformed incorrect options into correct ones aligned with the question's intent. Additionally, some options ended up testing overlapping knowledge points. To address this, **all refined questions were thoroughly cross-referenced by human experts against authoritative guidelines to ensure their accuracy and quality.** This rigorous process minimizes bias in question generation and enhances the overall quality of the questions.
>
> We have included this statement in the revision in Section 3.2.
>
> Thank you once again for your valuable and insightful feedback. If there are any remaining questions or points of clarification, we would be more than happy to address them. We hope our responses have successfully resolved your concerns and would greatly appreciate it if you could reconsider and potentially adjust your score.
>
> Best regards,
>
> Authors of Submission 12083
>
> [5] Grassini, Simone. "Shaping the future of education: exploring the potential and consequences of AI and ChatGPT in educational settings." Education Sciences 13.7 (2023): 692.
>
> [6] Yu, Hao. "Reflection on whether Chat GPT should be banned by academia from the perspective of education and teaching." Frontiers in Psychology 14 (2023): 1181712.

---

> > ### Comment · Reviewer_t1YP · 2024-11-26
> > **response to authors**
> >
> > I thank the author for the detailed response. My concerns are partially addressed. I have a follow-up question. I am concerned that seeing the high-performance of evaluated LLMs, above 80%, that newer models such as the latest version of Claude-3.5-Sonnet and o1, may already saturate this benchmark. I wonder if the authors have any insight about this? this is important as it ls likely that safety knowledge is correlated with capabilities, such that once LLMs become used in laboratory environments, they may saturate this benchmark.

---

> ### Author Response · Authors · 2024-11-26
> **Newer Models are Hard to Saturate this Benchmark**
>
> ### **Question**: My concerns are partially addressed. I have a follow-up question. I am concerned that seeing the high-performance of evaluated LLMs, above 80%, that newer models such as the latest version of Claude-3.5-Sonnet and o1, may already saturate this benchmark. I wonder if the authors have any insight about this? this is important as it ls likely that safety knowledge is correlated with capabilities, such that once LLMs become used in laboratory environments, they may saturate this benchmark.
>
> ### **Response**:
> **Newer Models with Advanced Reasoning Ability Do not Saturate this Benchmark**
>
> We sincerely appreciate your feedback. In our experiments, we have evaluated the performance of the latest Claude-3.5-Sonnet on LabSafety Bench, and as shown in Table 1, its accuracy is slightly lower than that of GPT-4o. Additionally, we recently tested **o1-preview** and **o1-mini** on the text-only questions of LabSafety Bench (since o1 does not currently support image-based questions). The results show that **o1-preview** achieved only 84.34% accuracy, and **o1-mini** achieved 79.27%, both performing worse than GPT-4o and GPT-4o-mini. We analyzed this discrepancy and concluded that while o1 models demonstrate strong reasoning abilities, LabSafety questions are primarily knowledge-intensive tasks that require LLMs to have explicitly learned the relevant knowledge points during training and accurately apply them—an aspect that advanced reasoning alone cannot overcome. A closer inspection of o1’s errors revealed patterns similar to those observed with GPT-4o, with overlapping error cases. Common issues include incorrect prioritization of hazards, overfitting to specific scenarios, or occasional hallucinations leading to incorrect responses. These findings suggest that insufficient fine-grained training data might result in systematic errors on certain questions, which cannot be addressed by reasoning ability alone.
>
> **High Accuracy but Significant Room for Improvement in Lab Safety Ability**
>
> Moreover, as detailed in Appendix B.5, we utilized expert-annotated difficulty, discrimination, and guessing parameters to estimate the Item Characteristic Curve (ICC) of our dataset using **Item Response Theory (IRT)**. The ICC demonstrates how the **Probability of Correct Response** changes with the test-taker's ability. For reference, an ability level of 3 corresponds to nearly perfect performance on our benchmark. However, GPT-4o's accuracy corresponds to an ability level of just **1.24**, indicating that despite its relatively high accuracy, GPT-4o still has significant room for improvement before achieving near-perfect performance.

---

> > ### Comment · Reviewer_t1YP · 2024-12-01
> >
> > I thank the authors for the additional clarification. Upon careful consideration and reading the other reviews, I still have concerns about the significance and motivation of the benchmark and will keep the score.

---

### Official Review · Reviewer_DEwk · 2024-11-05

**Soundness:** 2
**Presentation:** 4
**Contribution:** 2
**Rating:** 3
**Confidence:** 4

**Summary:**

This paper considers the problem of an LLM being queried on laboratory protocols and inadvertently generating responses which cause users to take unsafe actions that cause laboratory accidents. To measure this risk, the paper introduces a benchmark of multiple-choice lab safety questions, similar to tests given in formal lab safety trainings, and compares the results to human baseline performance. The paper also compares a variety of commercial models including open-weight models and proprietary access models, exploring the nature of the observed differences in performance.

**Strengths:**

+ Benchmarks have a difficult epistemic relationship with risk of unwanted behavior. The validation study performed here is welcome and useful.
+ There is substantial engagement with the performance evaluation results, including (informal) qualitative review of errors and of lower-performing models to understand approaches for enhancing support.

**Weaknesses:**

- The paper's core material is substantially disconnected from its motivation, to the point of being potentially misleading. These claims need to be tempered. In specific, there is no offered model of how users reach unsafe actions. The model of "a user queries an LLM and then takes an action which might be unsafe" is naive to the point of straining credulity, and this is not even proposed except in a latent way. The motivating accidents offered between 036 and 040 are, in general, not lab accidents of the sort that improved LLM function would avoid. How does the LLM query fit into a process for operating things like chemical fabrication pipelines safely? Why would a human lab scientist with safety training unquestioningly follow guidance from an LLM? Why would a lab user without safety training be in the position to access materials that could lead to the kinds of issues motivating this work? Yes, personnel benefit from automated guidance and may act counter to safety protocols for a variety of reasons, but what is the threat model justifying this work (without which the value of the work is hard to establish or evaluate)? Relatedly, the approach of focusing on all manner of labs should be justified more cleanly from the beginning - it took me to almost the end of the paper to understand that the goal was to evaluate the performance of current models against these risks (again, because the risk model is not made explicit) rather than try to develop a benchmark which would aid the development of domain-specific models or of model tuning. At a systems level, it is unlikely that actions taken on the equivalent of the honor system can, on their own, cause an accident. It's possible, but the model needs to be clearer or the motivation and top level claims tempered accordingly.
- As the validation moves forward, the paper concludes that having certain structured knowledge could benefit the generation of safe advice, but there is surprisingly little study in the paper of methods for integrating symbolic representation or knowledge bases into generation. I am interested to know how much this can help, if at all.
- Tests against benchmarks do not define much if any knowledge about the avoidance of risk. The paper should caveat the measurement of safety against some notion of the claim that's actually required, which is that the model under test is fit for purpose for some defined purpose. It would be valuable, I think, to link not only to workflow tasks but to some kind of ontology of safety practices vs. lab tasks, which is essentially what the standards used for question generation provide. Why not maintain this structured representation? What is gained/lost by transitioning to this approach?
- I am somewhat concerned about the approach of using LLMs to (re)generate benchmark questions. In specific, if LLM text is detectable, can this rewriting of questions bias the selection of multiple-choice answers by models during benchmark evaluation? Why or why not?
- Although part of the evaluation used human subjects, I cannot find an ethics statement or other mention of human subjects research review.

Minor nit:
- Reference (OSHA, 2011l) is somewhat challenging to read due to the orthographic similarity of the 1 and the l. I'm not sure what to do about this, but perhaps there are ways to find alternative keys?

**Questions:**

* What is the model of safety risk against which this benchmark is meant to measure model performance? How can I turn the performance evaluation against the benchmark into a claim about whether certain actions or workflows will be safe? Should there be a process model for integrating LLM-driven actions into lab activities?
* What are the tasks against which models are being measured by this benchmark?

**Details Of Ethics Concerns:**

The paper reports the measurement of a human baseline on the provided benchmark, but contains no indication that the use of human subjects was reviewed or approved. Although this is very likely minimal-risk human-subjects research, I'm unsure about the forms of compliance required under ICLR policies.

---

> ### Author Response · Authors · 2024-11-21
> **Response to Weakness 1: Disconnect Between Motivation and Core Materials**
>
> Dear Reviewer DEwk,
>
> We sincerely thank you for your incredibly detailed review, as well as your thoughtful and in-depth suggestions and questions. In particular, your insights regarding how LLM-driven lab safety judgments can be integrated into laboratory systems have provided valuable perspectives. We have carefully reflected on these points and incorporated the additional content from our rebuttal into the revised manuscript.
>
> **Response to Weakness 1: Addressing the Perceived Disconnect Between Motivation and Core Materials**
>
> We are sorry to learn that the reviewer understood “ that the goal was to evaluate the performance of current models against these risks” “almost at the “end of the paper”. We believe we clearly outlined our goals throughout the paper.
>
> For example, **in the abstract**, we explicitly state:  “We propose the LabSafety Bench, a comprehensive evaluation framework….” and “Our evaluations demonstrate that while GPT-4o outperforms human participants,......”. **In the introduction**, we emphasize multiple times, “To address this question, a systematic evaluation of LLMs’ trustworthiness in the context of lab safety is essential”. “ To address this challenge, we propose a Laboratory Safety Benchmark (LabSafety Bench), a specialized evaluation framework designed to assess the reliability and safety awareness of LLMs in laboratory environments.”  “We evaluate the performance of 17 foundation models on LabSafety Bench, 7 open-weight LLMs, 4 open-weight VLMs, and 6 proprietary models. ”  **One of our summarized contributions at the end of the introduction** reiterates that “We conduct extensive evaluations of LLMs and VLMs using LabSafety Bench. Our findings show that GPT-4o achieves the highest accuracy on these questions.”
>
> Given these explicit statements in both the abstract and introduction, **we respectfully disagree with the reviewer's comment that “The paper's core material is substantially disconnected from its motivation.”** We are confident that we have clearly communicated the primary objective of evaluating current models against lab safety risks. However, we appreciate the feedback and would be very grateful for a discussion to better understand the specific concerns. We sincerely hope that, after considering our clarifications in the following, the reviewer might reconsider the evaluation of our work.
>
> We are glad that the reviewer now understands that our work is positioned as an evaluation rather than focusing on "try to develop a benchmark which would aid the development of domain-specific models or of model tuning" Starting from this clarified understanding, we would like to address the series of questions raised by the reviewer, which primarily center around clarifying the **context and scenarios in which the LabSafety Bench evaluations are relevant**, such as  "How does the LLM query fit into a process for operating things like chemical fabrication pipelines safely? Why would a human lab scientist with safety training unquestioningly follow guidance from an LLM? Why would a lab user without safety training be in the position to access materials that could lead to the kinds of issues motivating this work? what is the threat model justifying this work ?"
>
> Best regards,
>
> Authors of Submission 12083

---

> ### Author Response · Authors · 2024-11-21
> **Response to other questions in Weakness 1**
>
> ### **Question**: How does the LLM query fit into a lab process?
>
> **Response**: There are increasing applications of LLMs in laboratory workflows, serving to enhance efficiency and accuracy in various stages of lab operations. First, in **remote and autonomous self-driving lab settings**, LLMs have been demonstrated to assist in drafting these protocols, reducing planning time and effort [1,2,3,4].  Second, there is a notable increase in the use of LLMs by **students and researchers across varying levels of expertise** [5,6,7]. LLMs could act as accessible knowledge repositories, providing quick guidance on standard practices, concepts, and procedures.
>
> ###  **Question**: How can the involvement of LLMs pose risks to the lab, even when lab users are well-trained?
>
> **Response**: Please note that while lab safety training can prevent individuals from engaging in unsafe practices, laboratory accidents can still occur due to unforeseen circumstances, human error, or gaps in knowledge [8]. The broader management of lab safety regarding user training falls under a separate domain of responsibility. Our focus is specifically on examining the role of LLMs and their impact on safety within lab environments. In the above-mentioned scenarios where LLMs are integrated into workflows, users may be actively involved in decision-making or not involved at all in the case of fully automated processes. Our focus is not on how experts can intervene to correct errors but rather on understanding and addressing the inherent risks and limitations posed by LLMs themselves, irrespective of user expertise.
>
> ###  **Concern**: At a systems level, it is unlikely that actions taken on the equivalent of the honor system can, on their own, cause an accident.
>
> **Response**: If the plan provided by an LLM is unreliable, it is no different from a decision made by an inadequately trained novel researcher, potentially leading to dangerous outcomes. For example, in Figure 3, most models, including GPT-4o, failed to correctly identify the substance as picric acid. If a researcher relies on the LLM's incomplete guidance, even slight friction or impact could trigger an explosion.
>
> [1] Daniil A. Boiko, Robert MacKnight, Ben Kline and Gabe Gomes. Autonomous chemical research with large language models.  Nature. volume 624, pages 570–578 (2023).
>
> [2] Milad Abolhasani and Eugenia Kumacheva. The rise of self-driving labs in chemical and materials sciences. Nature Synthesis. volume 2, pages 483–492 (2023)
>
> [3] Latif, Ehsan, Ramviyas Parasuraman, and Xiaoming Zhai. "PhysicsAssistant: An LLM-Powered Interactive Learning Robot for Physics Lab Investigations." arXiv preprint arXiv:2403.18721 (2024).
>
> [4] Gary Tom et al. Self-Driving Laboratories for Chemistry and Materials Science. Chem. Rev. 2024, 124, 16, 9633–9732.
>
> [5] Sebastian Tassoti. Assessment of Students Use of Generative Artificial Intelligence: Prompting Strategies and Prompt Engineering in Chemistry Education. J. Chem. Educ. 2024, 101, 6, 2475–2482.
>
> [6] Meng-Lin Tsai ,  et al .  Exploring the use of large language models (LLMs) in chemical engineering education: Building core course problem models with Chat-GPT. Education for Chemical Engineers Volume 44, July 2023, Pages 71-95.
>
> [7]  Andy Extance. ChatGPT has entered the classroom: how LLMs could transform education.   Nature; London Vol. 623, Iss. 7987,  (Nov 16, 2023): 474-477. DOI:10.1038/d41586-023-03507-3.
>
> [8] Ménard, A. Dana, and John F. Trant. "A review and critique of academic lab safety research." Nature chemistry 12.1 (2020): 17-25.

---

> > ### Comment · Reviewer_DEwk · 2024-11-26
> >
> > I should perhaps have stated my concern more clearly: there is not a model in the paper for how LLMs will be used in the laboratory control workflow.
> >
> > Although the language is consistently broad throughout, one might see this as a negative rather than a positive: a research paper should be specific and precise in its claims, not broad and expansive (or at least not moreso than is justified by the work). The motivation is _extremely_ broad. Only at the end, in the conclusion, does the paper indicate in a precise way that the goal is to examine the situation of "advice given by current-generation models".
> >
> > A risk model could be scenario-based as described in the response. More specifically, though, I'd like to see the paper describe an actual lab setting and the way an LLM fits in the work practices of the lab workers. From there, one can extract real, meaningful safety requirements on advice. People can get bad advice from lots of places: colleagues in other labs who don't understand the specific context; textbooks; the web; outdated manuals or procedures; colleagues in the lab who are miscommunicating. I struggle to see how to interpret the "safety" claims about the benchmark without some notion of how to compare them to realistic risks. The claim here, in the response, about planning for autonomous labs is _much_ closer. But how does safety work in these contexts? Surely even a "self-driving" lab won't merely follow a protocol generated by an LLM without some sort of review or limitation. Limitations could be structural: lots of lab work involves repetitive work to sweep a parameter space to characterize either a setup or a sample or the properties of an application. This repetitive work is low risk, both in the sense that the outcomes are unsurprising to the researchers and in the sense that the autonomous components likely don't have access to any tools, techniques, or materials which could be recombined to create a dangerous situation.
> >
> > Overall, I still see a big gap between the proposed approach and the strength of the claims around safety. One way to fix this is to temper the claims appropriately: the benchmark evaluates the likelihood that the model will produce information consistent with known and accepted protocols. There are, as the review suggests, ways to expand this work towards the scope of the claims made in the paper, but a conference review is the wrong place to ask for a different study. Take these suggestions under advisement for future work!

---

> ### Author Response · Authors · 2024-11-21
> **Response to Weakness 2 and Weakness 3**
>
> ### **Weakness 2**: As the validation moves forward, the paper concludes that having certain structured knowledge could benefit the generation of safe advice, but there is surprisingly little study in the paper of methods for integrating symbolic representation or knowledge bases into generation. I am interested to know how much this can help, if at all.
>
> **Response**:
> We would love to discuss this question. However, we could not identify a specific section in our paper that makes such a claim "The paper concludes that having certain structured knowledge could benefit the generation of safe advice," we apologize if there was any misunderstanding. Could you please specify the part you are referring to?
>
> In our work, we primarily discuss the structured categorization of our dataset, dividing it into four main categories and ten subcategories to ensure comprehensive coverage of lab safety topics. Given the complexity of lab safety protocols and the lack of existing datasets, our approach focused on collecting relevant knowledge points from authoritative sources and expert input. These knowledge points were then used to generate benchmark questions. However, we did not delve into the use of knowledge bases or symbolic representation in this process, because this is completely irrelevant to our evaluation work. Therefore, we are surprised to see the comment “there is surprisingly little study in the paper of methods for integrating symbolic representation or knowledge bases into generation”.  “knowledge bases or symbolic representation” could be helpful to improve LLM generation regarding lab safety. However, it is irrelevant to our evaluation work, as we are not yet in the stage of addressing these risks.   Of course, we could extend our future work to discuss the possibility of integrating the knowledge bases or symbolic representation for improving the LLM generation in the context of lab-related applications.
>
> We hope our explanation has clarified any misunderstandings about our paper. If you still have questions regarding our motivation, we would be more than happy to engage in a more in-depth discussion with you.
>
>
> ### **Weakness 3.1** Tests against benchmarks do not define much if any knowledge about the avoidance of risk. **and Question 1.2**: How can I turn the performance evaluation against the benchmark into a claim about whether certain actions or workflows will be safe?
>
> **Response**: Regarding the “knowledge about the avoidance of risk” or “whether certain actions or workflows will be safe”, we believe this is very interesting to study. Thanks for bringing this to our attention. This could be our next investigation regarding the enhancement of lab safety.  At least from our current study, GPT-4o is capable of understanding certain lab safety knowledge points. We could extend the research on the evaluation cases that GPT-4o succeeded, and vary the problem description for investigating the avoidance of risk.
>
>
> ### **Weakness 3.2**  The paper should caveat the measurement of safety against some notion of the claim that's actually required, which is that the model under test is fit for purpose for some defined purpose. It would be valuable, I think, to link not only to workflow tasks but to some kind of ontology of safety practices vs. lab tasks, which is essentially what the standards used for question generation provide. Why not maintain this structured representation? What is gained/lost by transitioning to this approach?
>
>
> **Response**: It is a great suggestion to generate questions based on “workflow tasks or some kinds of ontology of safety practices”. These are especially useful when generating advanced questions that involve a sequence or a series of operations, where risks could emerge across different steps in a sequence of operations. However, as we mentioned on line 90 and line 116 in Section 1 of our paper (line 99 and line 125 in the revision), our current evaluation primarily focuses on the “awareness” of risks. This is why we utilize knowledge points that are directly related to identifying potential hazards. However, based on our interpretation, we understand it as a recommendation to clarify the specific knowledge points or risks that we are evaluating, as well as the context in which these evaluations are meaningful.
>
> To address this, we enhanced the paper by adding the following statement in Section 3: “Since our study focuses on evaluating the overall lab safety awareness of LLMs, we opted not to assess them through workflow tasks or ontology-based safety protocols. Instead, our evaluation is grounded in knowledge points, providing a comprehensive measure of the model's understanding across various aspects of lab safety.”

---

> > ### Comment · Reviewer_DEwk · 2024-11-26
> >
> > I think we are talking past each other on the term "structured knowledge". I agree strongly that the construction of the data set is structured and that the evaluation test questions are structured. But also: the data set is (as I understand it) textual, meaning that the specific lab workflows and knowledge points are not structured. That's OK! LLMs show us that inferring structure is broadly valuable. Your question here is whether it is specifically valuable in a precise way, for which you really need to explore the question of "what kinds of risks exist?" Or, since that question is epistemically problematic, you might explore a closed-end question like "what kinds of accepted procedures exist?" (this is, in any case, what the tests evaluate). So I disagree strongly that comparing the benchmark to some kind of structured risk taxonomy is "completely irrelevant" - it is critically important! And indeed the work does "propose a new taxonomy for lab safety to ensure comprehensive coverage" (184-185). What is not explained is the level of detail in this taxonomy or how this claim of comprehensiveness is validated. Again, a conference review is not the right place to ask for different work, but could you somehow scaffold this knowledge such that the LLM generations become more reliable as advice in a concrete lab application setting? Without a clear sense of the way the benchmark aids claims of sufficiency in evaluation, it's hard to know if the benchmark is improving safety or just adding on to the evaluation. On the question of "where is this claim made?", it's broadly the second contribution bullet at the end of Section 1 (116-119) - a "benchmark for evaluating [...] lab safety awareness" should measure something about lab safety awareness. Either (a) don't claim to measure this; or (b) show you are measuring this. Again, the risk model is extremely relevant and the goals for use of the LLM generations need to be specified.
> >
> > To put this all slightly differently: when the tests against which you are comparing are created, experts map the desired knowledge points back to concrete lists of hazards (derived from real mishaps!). Replacing that with a bigger but less structured bank of questions _could_ be useful in showing that LLM outputs can be used safely in some workflow, but the question bank is not a direct replacement for the curated safety evaluation tests. Without a model of workflows or concrete risks, I struggle to see how you draw this comparison and show that the knowledge constructs you intend to capture through question answering are meaningfully the same.

---

> > > ### Author Response · Authors · 2024-12-01
> > > **Rebuttal to “The language is consistently broad throughout”**
> > >
> > > We sincerely thank the reviewer for their detailed and thoughtful feedback. We recognize the effort that went into providing such an in-depth response, and we appreciate the time and consideration given to our work. However, we respectfully disagree with certain aspects of the review, particularly regarding the characterization of our language and the interpretation of our contributions.
> > >
> > > ### **1. Rebuttal to “The language is consistently broad throughout”**
> > >
> > > Firstly, our paper does not exhibit the characteristic described as “the language is consistently broad throughout.” As stated in our initial rebuttal, we have repeatedly emphasized that our goal is to evaluate the performance of current models against the risks associated with lab safety. Indeed, our work focuses on providing a detailed evaluation of **LLM lab safety awareness**. The reviewer also mentioned that the claim, “the paper concludes that having certain structured knowledge could benefit the generation of safe advice,” originates from the end of Section 1 (lines 116-119). However, the exact sentence in our paper is:
> > >
> > > > “We introduce the first benchmark for evaluating foundational models in lab safety awareness issues. Under the guidance of a new taxonomy, we curate a wide range of relevant questions, ensuring their high quality through verification by human experts.”
> > >
> > > From this statement, we fail to see how it implies the conclusion attributed by the reviewer. While we recognize and appreciate the reviewer’s careful reading and thoughtful comments, we suspect there might have been an initial misinterpretation of our intent. This may have led to a preconceived notion that our paper aimed to show “having certain structured knowledge could benefit the generation of safe advice,” which we never claimed. Consequently, this misunderstanding might have contributed to the impression that our paper overstates its contributions or is overly broad, which we respectfully assert is not the case.

---

> > > ### Author Response · Authors · 2024-12-01
> > > **On Matching Evaluation to Real-World Applications**
> > >
> > > We understand the reviewer’s desire for an evaluation more closely aligned with real-world applications. However, it is important to clarify two key points regarding lab safety evaluations:
> > >
> > > 1. **Safety accidents result from failures in one or more safety protocols.** We acknowledge the complexity of real laboratory environments; however, every lab safety hazard, **no matter how intricate, ultimately arises from a combination of fundamental safety knowledge points**. For example, the ethanol pipeline explosion mentioned in the introduction highlights the core principle that flammable liquids must be handled to avoid ignition from static electricity during transfer. In our benchmark, this specific hazard is addressed through multiple questions targeting the handling of flammable substances. By evaluating LLMs’ understanding of individual safety knowledge points, we aim to ensure comprehensive coverage of critical lab safety protocols.
> > >
> > > 2. **Scenario-based evaluations can introduce bias.** Certain safety hazards, such as risks from liquid splashes, occur across a wide range of scenarios, while others, such as those in vacuum environments, appear far less frequently. Evaluating by scenario could result in disproportionately assessing LLMs on common hazards while underrepresenting their awareness of rarer, yet equally critical, safety hazards.
> > >
> > > 3. **Why We Do Not Directly Evaluate “What Kinds of Accepted Procedures Exist?”**
> > > Lab safety is a knowledge-intensive task, and scenario-based evaluations of acceptable procedures would introduce numerous branching paths for each step. Across an entire dataset, this would amplify the biases mentioned earlier, where common hazards dominate, and rare hazards are underrepresented.
> > > Instead, our approach begins with knowledge points to generate scenarios that require flexible application of these points. This allows us to evaluate whether LLMs can effectively apply their knowledge in diverse situations without being skewed by scenario-specific biases.
> > >
> > > 4. **Classification and Taxonomy Design**
> > > Our classification system is derived from authoritative guidelines, specifically OSHA [1], and further refined with input from the Risk Management and Safety team at a large university. This collaborative process ensures that the taxonomy is both professional and comprehensive. We will add this clarification to our revised manuscript to reinforce the credibility of our approach.
> > >
> > > 5. **Why We Do Not Use Real-World Accidents for Evaluation**
> > > We chose not to rely on real-world accidents as the basis for our evaluation because, as highlighted in [2], lab safety awareness in practice often focuses narrowly on recent incidents. For instance, after an accident caused by Factor A, significant attention is directed toward preventing similar incidents, only for another accident caused by Factor B to occur later. This piecemeal approach fails to address lab safety comprehensively, leaving gaps that could lead to future accidents. By evaluating LLMs on a broader set of knowledge points, our benchmark aims to assess lab safety awareness holistically rather than through the lens of isolated events.
> > >
> > > To address these concerns, we will clarify the above discussion in the revision.
> > >
> > > [1] OSHA. Laboratory safety guidance. URL https://www.osha.gov/sites/ default/files/publications/OSHA3404laboratory-safety-guidance. Pdf

---

> ### Author Response · Authors · 2024-11-21
> **Response to Weakness 4-6 & Question 1 and 2**
>
> ### **Weakness 4**: I am somewhat concerned about the approach of using LLMs to (re)generate benchmark questions. In specific, if LLM text is detectable, can this rewriting of questions bias the selection of multiple-choice answers by models during benchmark evaluation? Why or why not?
>
> **Response**: We would like to better understand your concern regarding the connection between "LLM-generated text being detectable" and "rewriting questions biasing the selection of multiple-choice answers by models during benchmark evaluation." We don't think these two are relevant. Specifically, are you suggesting that the model under evaluation might favor answers generated by an LLM? If so, we would like to clarify that all answer options were generated by LLMs, not just the correct ones.
>
> Alternatively, are you implying that the knowledge point might have been inherently difficult initially, but the question generated became more understandable to GPT-4o after the generation? If this is the case, we are here to explain why the generated questions will not be more understandable to GPT-4o. After refinement, some modified options were found to be inaccurately phrased, irrelevant to the question, or transformed incorrect options into correct ones aligned with the question's intent. Additionally, some options ended up testing overlapping knowledge points. To address this, all refined questions were **thoroughly cross-referenced by human experts against authoritative guidelines to ensure their accuracy and quality**. This rigorous process minimizes bias in question generation and enhances the overall quality of the questions.
>
> We have included this statement in the revision in Section 3.2.
>
> ### **Weakness 5**: Although part of the evaluation used human subjects, I cannot find an ethics statement or other mention of human subjects research review.
>
> **Response**:  **Please refer to Appendix C.2**, where we state: “The survey was approved by the Institutional Review Board (IRB) committee at the university, ensuring that all research involving human participants adheres to ethical guidelines and standards for privacy, consent, and safety.”
>
>
> ### **Weakness 6**: Reference (OSHA, 2011l) is somewhat challenging to read due to the orthographic similarity of the 1 and the l. I'm not sure what to do about this, but perhaps there are ways to find alternative keys?
>
>
> **Response**: Thank you for the feedback. We adjusted the citation formatting to make it clearer. Specifically, by specifying OSHA protocols as either OSHA Fact Sheets, OSHA Quick Facts, or General OSHA Protocols, we effectively addressed this issue.
>
> ### **Question 1.3**: Should there be a process model for integrating LLM-driven actions into lab activities?
>
> **Response**: Integrating LLMs into laboratory workflows is a promising but complex direction. We could definitely include this in the future work discussion. Unfortunately, addressing this process model is out of the scope of our current study.
>
> ### **Question 2**: What are the tasks against which models are being measured by this benchmark?
>
> **Response**: The LabSafety Bench evaluates all LLM models across tasks and assesses their understanding, reasoning, and application of lab safety knowledge. For VLM models, the model further evaluates the task of image understanding on Lab safety-related images.
>
>
> Thank you once again for your thoughtful and constructive feedback. If there is anything unclear or if you have further questions, we would be more than happy to address them. We hope our responses have effectively resolved your concerns, and if so, we kindly ask you to consider revisiting and possibly improving your score.
>
> Best regards,
>
> Authors of Submission 12083

---

### Official Review · Reviewer_nGEs · 2024-11-09

**Soundness:** 2
**Presentation:** 3
**Contribution:** 2
**Rating:** 6
**Confidence:** 4

**Summary:**

Laboratory accidents pose significant risks to human life and property, and despite safety training, unknowingly unsafe practices may occur. LLMs are increasingly relying on for guidance, raising concerns about their reliability in safety-related decision-making. The Laboratory Safety Benchmark (LabSafety Bench) is proposed to address this gap, assessing LLM and large vision models' performance in lab safety contexts. The benchmark includes 765 multiple-choice questions verified by human experts.

**Strengths:**

+ By focusing on laboratory safety, the paper tackles a highly specific yet essential aspect of LLM application. Unlike general benchmarks, the LabSafety Bench targets a field where errors can have severe, real-world consequences, filling a critical gap in LLM evaluation.
+ The creation of the Laboratory Safety Benchmark (LabSafety Bench) tailored to lab safety demonstrates a novel and practical contribution.

**Weaknesses:**

-  While multiple-choice questions allow for standardized assessment, they may not fully capture complex decision-making required in lab safety.
- How the human experts are selected and the examination procedure should be described in details.
- Some related works are missing, e.g.,

	- Xie, Tinghao, Xiangyu Qi, Yi Zeng, Yangsibo Huang, Udari Madhushani Sehwag, Kaixuan Huang, Luxi He et al. "Sorry-bench: Systematically evaluating large language model safety refusal behaviors." arXiv preprint arXiv:2406.14598 (2024).
	- Li, Kenneth, Oam Patel, Fernanda Viégas, Hanspeter Pfister, and Martin Wattenberg. "Inference-time intervention: Eliciting truthful answers from a language model." Advances in Neural Information Processing Systems 36 (2024).
	- Xie, Xuan, Jiayang Song, Zhehua Zhou, Yuheng Huang, Da Song, and Lei Ma. "Online Safety Analysis for LLMs: a Benchmark, an Assessment, and a Path Forward." arXiv preprint arXiv:2404.08517 (2024).

**Questions:**

1. What are the tasks and roles LLMs play in Lab environment? Why is it important to benchmark the safety? It should be highlighted in the intro.

2. The selection of human experts and the manual processing procedure is not so clear. It should be illustrated for readers' understanding the soundeness of your benchmark.

---

> ### Author Response · Authors · 2024-11-21
> **The rationale for using multiple-choice questions and the process for selecting and involving human experts**
>
> Dear Reviewer nGEs,
>
> We sincerely thank you for your meticulous review and the highly insightful suggestions and questions. Your feedback, particularly regarding the process of selecting human experts and the expert review of the questions, has been invaluable. We have incorporated the details discussed in our rebuttal into the paper to address these points and enhance its clarity and completeness. Thank you again for your thoughtful and constructive comments.
>
> **Weakness 1**: While multiple-choice questions allow for standardized assessment, they may not fully capture the complex decision-making required in lab safety.
>
> **Response**: In this pioneering study on lab safety, we chose to use multiple-choice questions as they support the usage of **standardized and quantifiable metrics like accuracy, making it feasible to compare the performance of different LLMs**. This is similar to the initial studies in MMLU [1], which first proposed benchmarks for Massive Multitask Language Understanding. It uses multiple-choice questions to facilitate the evaluation.
>
> We fully agree that multiple-choice questions alone cannot comprehensively capture the risks associated with lab safety. To address this limitation, we are actively working on exploring open-ended scenarios and formats to expand the evaluation dimensions. However, despite the trade-off in format for the sake of measurability, our questions are designed to **cover a broad range of knowledge domains**. These are grounded in **authoritative sources** such as WHO [2] and OSHA [3]. Moreover, the questions **span various difficulty levels** and **encompass biology, chemistry, and physics disciplines**. Although these questions are “not complex”, there are some notable gaps between different LLMs, and even the best LLM (GPT-4o) only scored 86.27% on both text-only and text-with-image questions in LabSafety Bench and 82.35% on hard questions, demonstrating that this benchmark is still challenging to current LLMs.
> Importantly, all questions have been **reviewed and refined by human experts** to ensure that **the options are non-trivial and effectively test the LLMs' understanding of the underlying knowledge points**.
>
>
> **Weakness 2**: How the human experts are selected and the examination procedure should be described in detail. **and**
> **Question 2**: The selection of human experts and the manual processing procedure is not so clear. It should be illustrated for readers' understanding the soundness of your benchmark.
>
> **Response**: Thank you for the question. We have included the following explanation in Appendix A.1, A.2, and A.3, regarding the expert selection, knowledge points collection, and human review process.
>
> **1. Selection of Experts with Relevant Subject Expertise**
> Human experts were selected from a large research university, targeting individuals with extensive experience in lab safety. We selected individuals with advanced educational backgrounds (PhD students or postdoctoral researchers) and at least three years of direct laboratory experience. Their expertise ensured a solid understanding of both theoretical and practical aspects of lab safety. For physics, biology, and chemistry, we selected 3 human experts respectively to review the questions.
>
> **2. Examination Procedure**
> **Knowledge Point Identification**: Experts began by identifying key lab safety knowledge points from authoritative sources, such as OSHA [2] and WHO [3]. These knowledge points formed the foundation for question development. Based on the knowledge points, GPT-4o helped generate and refine the questions.
>
> **Question Verification**: After refinement, some modified options were found to be inaccurately phrased, irrelevant to the question, or transformed incorrect options into correct ones aligned with the question's intent. Additionally, some options ended up testing overlapping knowledge points. To address this, all refined questions were thoroughly cross-referenced by human experts against authoritative guidelines to ensure their accuracy and quality. This rigorous process minimizes bias in question generation and enhances the overall quality of the questions. Each expert will review all the questions about the corresponding subject individually.
>
> **Panel Review**: Each question underwent a panel review by three subject-matter experts, who collaboratively evaluated its accuracy, difficulty level, and ability to effectively assess an LLM’s understanding of the corresponding knowledge. This process included detailed discussions to ensure consensus on the correct answer and the plausibility of the distractors.
>
> [1] Hendrycks, Dan, et al. "Measuring massive multitask language understanding." arXiv preprint arXiv:2009.03300 (2020).
>
> [2] WHO.Laboratorybiosafetymanual.WHO,5(2):1–109,2003.
>
> [3] OSHA. Laboratory safety guidance. URL https://www.osha.gov/sites/ default/files/publications/OSHA3404laboratory-safety-guidance. Pdf
>
> Best regards,
>
> Authors of Submission 12083

---

> ### Author Response · Authors · 2024-11-21
> **Suggested related works, the tasks and roles LLMs play in Lab environment and the significance of LabSafety Bench**
>
> **Weakness 3**: Suggested related works
>
> **Response**: We have cited and discussed these papers in the revised Section 2. Specifically, Sorry-bench [4] evaluates the degree of safety refusal by LLMs when faced with user queries involving unsafe questions, focusing on the misuse of LLMs.  The work of Li et al [5] improves the truthfulness of model outputs by shifting activations in the “truthful” direction during inference. The study in [6] conducts a comprehensive evaluation of existing online safety analysis methods for LLMs.
>
> However, these works primarily focus on traditional LLM trustworthiness, which is often limited to software-level issues such as truthfulness and safety. In contrast, our work focuses on evaluating LLM trustworthiness in lab safety, a domain involving **real-world physical safety risks**. While an LLM’s response may appear correct, its implementation in real-world scenarios could pose significant threats to personal and property safety. This real-world focus distinguishes our contribution and highlights the critical nature of our benchmark.
>
> **Question 1**: What are the tasks and roles LLMs play in Lab environment? Why is it important to benchmark the safety? It should be highlighted in the intro.
>
> **Response**: Thanks for the question.
>
> As discussed on Page 1, lines 43-47, the tasks and roles LLMs play in lab environments include assisting novice researchers with lab experiment procedures and supporting decision-making in self-driving labs. The importance of benchmarking LLM Safety can be found in lines 47-53 (lines 51-66, 73-74 in the revision).
>
> To further emphasize the roles of LLMs and the importance of evaluating their performance in lab safety, we have revised our introduction to include the following:
>
> LLMs can assist with **record-keeping, report writing, and data analysis**, as demonstrated by applications like LabTwin [7]. These capabilities enhance overall productivity but demand high accuracy to mitigate risks. Furthermore, LLMs can act as accessible resources for **addressing safety-related questions**, particularly **during time-sensitive tasks or when researchers are working alone in the lab**. In addition, in future autonomous or remote labs, LLMs could take on critical roles in **monitoring experimental progress, detecting hazards, and providing proactive safety recommendations**. Given the increasing usage of LLMs in this context, benchmarking LLM safety in the lab environment is an emergent requirement.  Our proposed benchmarking provides a structured framework to **evaluate whether these models can be trusted to make accurate and sound decisions under these conditions**. In addition, benchmarking LLMs evaluates the possibility of leveraging LLMs to assist in the lab safety training of students. Students often fail to retain all critical safety knowledge during training, which in turn frequently leads to laboratory accidents [8]. LLMs can act as supplementary resources to assist student training.
>
> Thank you again for your valuable and constructive feedback.  If any aspects remain unclear or if you have additional questions, we are more than happy to provide further clarification. We hope that our responses have addressed your concerns effectively, and we kindly ask you to reconsider and, if possible, update your score to reflect this resolution.
>
> Best regards,
>
> Authors of Submission 12083
>
> [4] Xie, Tinghao, Xiangyu Qi, Yi Zeng, Yangsibo Huang, Udari Madhushani Sehwag, Kaixuan Huang, Luxi He et al. "Sorry-bench: Systematically evaluating large language model safety refusal behaviors." arXiv preprint arXiv:2406.14598 (2024).
>
> [5] Li, Kenneth, Oam Patel, Fernanda Viégas, Hanspeter Pfister, and Martin Wattenberg. "Inference-time intervention: Eliciting truthful answers from a language model." Advances in Neural Information Processing Systems 36 (2024).
>
> [6] Xie, Xuan, Jiayang Song, Zhehua Zhou, Yuheng Huang, Da Song, and Lei Ma. "Online Safety Analysis for LLMs: a Benchmark, an Assessment, and a Path Forward." arXiv preprint arXiv:2404.08517 (2024).
>
> [7] Sukanija Thillainadarasan (Suki. “Real-Time Support to Scientists in the Lab by Leveraging LLM’s.” Labtwin.com, 2 Aug. 2023, www.labtwin.com/resources/real-time-support-to-scientists-in-the-lab-by-leveraging-llms, https://doi.org/105957269002/1687754136110/Labtwin-2023Mar.
>
> [8] Ménard, A. Dana, and John F. Trant. "A review and critique of academic lab safety research." Nature chemistry 12.1 (2020): 17-25.

---

### Author Response · Authors · 2024-11-21
**Global Response and Changes in the Revision**

Dear All Reviewers and Area Chairs,

We would like to express our heartfelt gratitude to all reviewers for their thorough evaluation and invaluable suggestions. Below, we summarize the main concerns and recommendations raised by the reviewers and outline the modifications we made in the manuscript to address them:

### **Rationale for Using Multiple-Choice Questions (Reviewers nGEs and GBeA)**
In response to concerns about using multiple-choice questions for evaluating lab safety, we clarified that they offer a **standardized and quantifiable metric**, enabling fair comparison across LLMs. While they may not fully capture complex decision-making, our **authoritative knowledge point collection** and **rigorous expert review** ensure the questions are challenging and test genuine understanding of lab safety concepts.

### **Selection of Human Experts and Review Procedures (Reviewer nGEs)**
We expanded on how human experts were selected and their review process. All experts are **experienced professionals with substantial lab experience**, and each question underwent **panel review** to ensure accuracy and difficulty. Details are provided in **Appendix A.1, A.2, and A.3**.

### **LLMs’ Roles and Tasks in Lab Environments (Reviewers nGEs, DEwk, and t1YP)**
In **Section 1**, we clarified LLM roles, including **assisting novice researchers** and **supporting decision-making in self-driving labs**. Additionally, examples like LabTwin show their use in **real-time record-keeping and information retrieval**, though safety concerns limit their application to experiment-related tasks. This discussion was added in Section 1 to better contextualize LLMs' significance in labs.

### **Significance of the LabSafety Bench (Reviewers nGEs and t1YP)**
In **lines 47-53**, we highlighted LabSafety Bench's role in evaluating LLM reliability in lab environments, crucial in labs where trustworthiness is key. We noted that **students, even after safety training, often face lab accidents**, and LabSafety Bench can assess LLMs' potential to **enhance safety training** and support safe operations. We highlight the significance of LabSafety Bench in Section 1.


### **Suggestion of Using Workflow Tasks or Ontology-Based Protocols for Benchmark Design (Reviewer DEwk)**
We chose a **knowledge-point-based evaluation** for its comprehensive assessment of LLMs’ understanding, which other approaches like workflow tasks or ontology protocols cannot match in depth and coverage. This is further clarified in **Section 3**.

### **Potential Bias in GPT-4o-Generated Questions (Reviewers DEwk and t1YP)**
To address concerns about GPT-4o biases, we emphasized the **human expert review process**, which corrected inaccuracies and errors, significantly reducing bias. Details are added in **Section 3.2**.

We hope these modifications address the reviewers’ concerns and demonstrate the rigor and reliability of our benchmark. Once again, we sincerely thank the reviewers for their insightful feedback, which has greatly enhanced the quality and clarity of our work.

Best regards,

Authors of Submission 12083

---

### Meta-Review · Area_Chair_3aKT · 2024-12-19

**Metareview:**

This paper introduces LabSafety Bench, a benchmark designed to evaluate the capabilities of LLMs in understanding and providing guidance on laboratory safety protocols. The work tackles an important topic, particularly as LLMs are increasingly integrated into scientific workflows. However, the paper has notable shortcomings, particularly in the process of establishing the benchmark.

**Additional Comments On Reviewer Discussion:**

The AC noted that one of the reviewers did not provide any responses during the rebuttal phase, so the AC personally reviewed the paper while making the decision. Unfortunately, the AC still has concerns regarding the paper, with the core issues centered around the process of establishing the benchmark. Specifically, under the lab safety problem, the saturation of the benchmark’s use cases and the rationality of the pipeline represent key shortcomings of the work, which aligns with the reviewers’ concerns. Considering the significantly lower scores of this paper compared to the acceptance threshold, the decision has been made to reject the submission.

---

### Decision · Program_Chairs · 2025-01-22

Reject